# Effect of nanoencapsulation on the solubility and antioxidant activity of astaxanthin pigmented oil extracted from shrimp waste (*Litopeneaus vannamei*)

Renata Nayane Fernandes dos Santos[1], Thaís Souza Passos[1,2], Rafael da Silva Fernandes[3], Kátia Nicolau Matsui[4], Francisco Canindé de Sousa Júnior[1,5], Karla Suzanne Florentino da Silva Chaves Damasceno[1,2], Cristiane Fernandes de Assis[1,5]*

1 Postgraduate Program in Nutrition, Center of Health Sciences, Federal University of Rio Grande do Norte, Natal, Rio Grande do Norte, Brazil, 2 Department of Nutrition, Center of Health Sciences, Federal University of Rio Grande do Norte, Natal, Rio Grande do Norte, Brazil, 3 Institute of Chemistry, Federal University of Rio Grande do Norte, Natal, Rio Grande do Norte, Brazil, 4 Departament of Chemical Engineering, Center of Technology, Federal University of Rio Grande do Norte, Natal, Rio Grande do Norte, Brazil, 5 Department of Pharmacy, Center of Health Sciences, Federal University of Rio Grande do Norte, Natal, Rio Grande do Norte, Brazil

* cristiane.assis@ufrn.br

## Abstract

Astaxanthin-pigmented oil from shrimp waste meal was nanoencapsulated by O/W emulsification using porcine gelatin (EAG) and a combination with soy protein (EAGS 2:2 and EAGS 3:1) to improve the solubility and antioxidant activity of the pigmented oil. The encapsulates presented spherical shape and smooth surface; particle size equal to 159.68 (14.42) nm for EAG, 192.72 (10.44) nm for EAGS 2:2, and 95.41 (17.83) nm for EAGS 3:1; amorphous structure; and chemical interactions. The oil incorporation efficiency ranged from 87.60–89.20%, the percentage of astaxanthin incorporated was approximately 68%, and the dispersibility in water around 50%. The antioxidant potential evaluation indicated that all formulations preserve or enhance the antioxidant activity of the oil up to three times than non-encapsulated oil. Therefore, porcine gelatin alone or in combination with soy protein was effective in promoting the solubility and enhancing the antioxidant activity of the astaxanthin-pigmented oil, demonstrating interesting characteristics for use in food.

## Introduction

Shrimp, as a fishing product, has a high market value and is commonly sold in processed form, peeled and without the cephalothorax, a rigid segment covered by a hardened exoskeleton containing chitin, which represents around 50 to 60% of the crustacean's body weight. As a result, shrimp processing generates a large amount of waste, which is disposed of inappropriately [1, 2].

On the other hand, it is known that with adequate knowledge and technology for the exploitation of raw materials, the waste can be reused, reducing the impact on the environment. Furthermore, when considering the nutritional value of shrimp processing waste, the

**Data Availability Statement:** All relevant data are within the manuscript and its Supporting Information file.

**Funding:** This study was partly financed by the Coordenação de Aperfeiçoamento de Pessoal de Nível Superior - Brasil (CAPES) - Finance Code 001.

**Competing interests:** No authors have competing interests.

presence of proteins, lipids, minerals, chitin, and sources of carotenoids, reuse can also contribute to the formulation of new ingredients and products for the food industry [3].

The primary carotenoid astaxanthin (3,3'-dihydroxy-β,β-carotene-4,4'-dione), an orange-red oxycarotenoid pigment, is found in the cephalothorax of shrimp in free form, conjugated with protein or esterified with one or more fatty acids [4–6]. Shrimp cannot synthesize astaxanthin but obtain it by ingesting a diet rich in algae or supplementing the feed [7].

Astaxanthin is attributed to some beneficial effects on health, such as anticarcinogenic, anti-diabetic, and anti-inflammatory activity, and a protective effect against cardiovascular diseases [8]. The natural pigment promotes protection against oxidative damage due to hydroxyls in its structure, which neutralize free radicals [9]. This antioxidant activity of astaxanthin is known to be greater than β-carotene and α-tocopherol [10].

The astaxanthin molecule is a lipophilic compound that can be extracted using hydrophobic solvents or vegetable oils [11]. A previous study by our research group Silva et al. (2021) [11] used flour obtained from shrimp waste to obtain oil pigmented with astaxanthin, aiming to formulate new food products. During the extraction process, the pigment becomes unstable due to several factors, such as the presence of light, oxygen, acidity, and heat, which causes the loss of its functional properties [7, 10, 12, 13].

In this sense, searching for new technologies that promote the preservation of the astaxanthin molecule has become necessary. One of our research group's strategies is nanoencapsulation, which effectively promotes the protection of bioactive compounds in vegetable oils [14–16]. The encapsulating agents protect the bioactive molecules from oxidation and decomposition (humidity, light, heat) [17]. Furthermore, nanoencapsulation can also promote increased water solubility, bioaccessibility and bioavailability, and enhancement of functional properties such as antioxidant activity, requiring a smaller amount of active ingredient to promote the desired effect [18, 19].

Among the techniques used to encapsulate lipophilic materials, oil-in-water (O/W) emulsification stands out. This technique is characterized by the dispersion of the oily phase in an aqueous phase with the aid of a surfactant to promote stability in the system [19]. Furthermore, after encapsulation, a drying method (freeze-drying and spray-drying) can be used to maintain the molecules' stability, allow rehydration when necessary, and expand the possibilities of use in food products [20].

The choice of encapsulating agent is also crucial to promote system stability. In this sense, proteins are commonly used as wall material due to their amphiphilic characteristics, emulsifying properties, and ability to form gels and viscoelastic films, promoting physical stability [21]. Gelatin is an animal biopolymer derived from collagen from different animal sources (porcine, bovine, fish). It is non-toxic, water-soluble, and biodegradable and has demonstrated the good formation of gels and films with interesting emulsifying properties for use in food. In addition, it is considered an excellent support for oils to preserve quality and increase shelf life [22]. Isolated soy protein is attractive due to its nutritional content and surface properties that allow adsorption at O/W interfaces, forming a protective layer that reduces surface tension [23].

Given the above, this work aims to promote the encapsulation of astaxanthin-pigmented oil using porcine gelatin and soy protein, isolated and combined, aiming at improving dispersibility in water and preserving and/or enhancing antioxidant activity, in addition to the sustainable alternative of using shrimp waste for future use in food.

## Materials and methods

### Materials

*Litopeneaus vannamei* shrimp residue (cephalothorax) was obtained from the shrimp processing company Faif's mariculture in Parnamirim, Rio Grande do Norte (RN), Brazil. Soy protein

isolate was donated by SUPRO® 620 IP Solae. Porcine gelatin (Type A) and Tween 20 were purchased from Sigma-Aldrich®, and soybean oil (Soya®) was purchased from a local market in Natal, RN, Brazil.

**Obtaining the pigmented oil with astaxanthin.** The residue was dried for 8 hours in a ventilated oven (Lucadema) at 70˚, then crushed in a food processor and domestic blender (Phillips Walita Mod. 6000W) and sieved (Bertel 10 MESH). The methodology of Silva et al. (2021) [11] was used to obtain the pigmented oil, in which the residue flour (100 g) was homogenized in soybean oil (400 mL) and subjected to heating without stirring (Tecnal TE 055) for 2 h at 70˚. Then, filtration was performed with gauze and subsequent centrifugation (Solab SL 760) at 3000 g at 25˚ for 10 minutes. The supernatant oil layer was separated with a pipette and later stored at -4˚ until use.

**Quantification of astaxanthin.** The pigmented oil sample was solubilized in acetonitrile at 0.1 mg/mL, filtered through a 0.22 μm membrane, and subjected to analysis by High-Performance Liquid Chromatography (HPLC).

The separation was performed according to Kim et al. (2013) [24] with modifications, using a Shim-Pack CLC-ODS (M) C18 reversed-phase column consisting of silica modified on the surface by octadecyl groups, with pores of 5.0 μm in diameter and column size of 25 cm x 4.6 mm in internal diameter, maintained at 40˚. The composition of the mobile phase was water (Phase A) and acetonitrile (Phase B), both containing 1% acetic acid, with a flow rate of 1.0 mL/minute. Elution was performed using a linear gradient from 0 to 30% B in 10 minutes, 30 to 70% B in 5 minutes, 70 to 100% in 10 minutes, and 100% B maintained for 5 minutes.

**Encapsulation of pigmented oil.** The particles were obtained using the O/W (oil in water) emulsification technique based on Castro et al. (2020) with modifications. The encapsulating agents used were isolated soy protein (EAS), porcine gelatin (Type A) (EAG), and a combination of the two in the ratios of 2:2 w/w and 3:1 w/w of porcine gelatin and isolated soy protein (EAGS 2:2 and EAGS 3:1), and Tween 20 as a surfactant.

For the oil phase, 10 mL of pigmented oil were used. Aqueous Phase 1 (FA 1–90 mL) consisted of 1.5% Tween 20 solubilized in distilled water, and Aqueous Phase 2 (FA 2–100 mL) consisted of 4% (w/v) of the encapsulating agent and 1.5% Tween 20 solubilized in distilled water. The gelatin was prepared separately under magnetic stirring (DiagTech DT3110H) for 1 hour at 40˚C. The soy protein and Tween 20 were solubilized under magnetic stirring (DiagTech DT3110H) for 30 minutes at room temperature. The formulations with combined encapsulating agents were homogenized under magnetic stirring (DiagTech DT3110H) for 1 hour at 30˚C at room temperature.

The emulsion was obtained by homogenizing the aqueous Phase 1 with the oil phase, using an ultradispersor (IKA ULTRA-TURRAX) at a speed of 17,000 RPM for 10 minutes. Subsequently, the homogenization of FA 2 with the obtained emulsion was performed using the same conditions described above. Finally, the obtained emulsions were lyophilized (LioTop L101) at -57˚ and at a pressure of 43 μHg.

## Characterization of nanoparticles

**Colorimetry.** The color evaluation was performed using a FRU® digital colorimeter, model wr10qc. The samples were dispersed on plates with a white background, exposed to the artificial light of 2000 Lux, and evaluated in triplicate at a distance of 4 cm. The color parameters were recorded on CIELAB (International Commission on Illumination) color scales that include $L^*$ [black (0) to white (100)], $a^*$ [green (-) to red (+)], and $b^*$ [blue (-) to yellow (+)] [25].

**Scanning Electron Microscopy (SEM).** The encapsulates were suspended in acetone and dripped onto silicon wafers fixed with carbon tape on stubs. The analysis was performed at

different magnifications using a ZEISS (AURIGA) SEM-FEG microscope, high vacuum, 2–3 kV voltage, and without metallization.

**Dynamic Light Scattering (DLS).** The mean diameter and polydispersity index were evaluated using the NANO-flex II 180˚ DLS Particles Size (Colloid Metrix) equipment. For the measurement, the encapsulates were cross-linked based on Castro et al. (2020) [14] with modifications, aiming to promote the deagglomeration of the particles. The measurements were performed in triplicate (60 seconds each measurement) using 2 mL of suspension. The data were analyzed using the NANO-flex Control 0.9.7 software. The experiment was performed in triplicate.

**Zeta potential.** The Zeta Potential of the encapsulated products was investigated in the pH range of 1 to 11 using the STABINO II Particle charge Titration equipment (Colloid Metrix). Thus, 10 mg of each formulation were diluted in 10 mL of ultrapure water ($\geq 18$ M$\Omega$ cm$^{-1}$) and then transferred to the cylindrical Teflon cell with the addition of aliquots (10 μL) of a strong acid (HCl– 0.1 N) or a strong base (NaOH– 0.025 N).

**Fourier Transform Infrared Spectroscopy (FTIR).** The raw materials (pigmented oil, porcine gelatin, soy protein, Tween 20) and the encapsulated materials were homogenized with potassium bromide (KBr), macerated and pressed to form tablets and, afterward, recorded in transmittance and with the mid-infrared region, from 400 to 4000 cm$^{-1}$. A Shimadzu spectrometer (FTIR-8400S, IRAFFINITY-1 series, IR SOLUTION version 1.60 software) was used with a scan number of 32 and resolution of 4 cm-1.

**X-ray diffraction (XRD).** The materials were placed in cylindrical sample holders and analyzed in X-ray diffractometers with a diffraction angle of 2θ between 0 and 100˚ in a high-resolution X-Ray diffractometer (SHIMADZU, XRD 7000 model) with Seifert generator ID3000.

**Incorporation efficiency.** The incorporation efficiency (%) of the pigmented oil into the particles was determined based on El-Messery et al. (2019) [26] with modifications. Thus, 15 mL of hexane were added to 1.5 g of each formulation, under agitation in a rotary incubator (QUIMIB—Q816M20) for 2 minutes, and then mixed on 150 mm Unifil quantitative filter paper. Then, the content retained on the filter paper was washed twice with 20 mL of hexane. The liquid phase was collected, and the hexane was removed in a rotary incubator (QUIMIB—Q816M20) at 30˚C for 48 h. The remaining oil represents the non-encapsulated fraction present on the surface of the particles. Calculating the difference between the initial amount of oil used to promote encapsulation and the free oil present on the surface of the particles makes it possible to obtain the amount of oil encapsulated, according to Eq 1.

$$OE(g) = OUE - OR \qquad (Eq 1)$$

OE: Oil encapsulated (g); OUE: Oil used on encapsulation (g); OR: oil remaining (g).

The percentage of astaxanthin-containing oil incorporated into the particles was obtained from Eq 2.

$$IE(\%) = \left[\frac{OE}{OI}\right] * 100 \qquad (Eq 2)$$

OE = Oil encapsulated (g), e OI = total pigmented oil used in the process (g).

In addition to determining the encapsulation efficiency of the pigmented oil, the incorporation efficiency of the astaxanthin present in it was also determined based on the evaluated content described above and considering the EI result obtained for the oil. Thus, the

percentage of astaxanthin present in the oil incorporated into the particles was obtained from Eq 3.

$$IE = \left[\frac{AE}{AI}\right] * 100 \qquad \text{(Eq 3)}$$

AE = astaxanthin present in the encapsulated pigmented oil (g), and AI = total astaxanthin present in the pigmented oil used in the process (g).

**Water dispersion test.** The determination was performed based on Paula et al., 2019 [27] and Silva et al., 2023 [28] with modifications. Thus, 100 mg of the encapsulated and soybean oil pigmented with astaxanthin were weighed in triplicate and incorporated into 5 mL of distilled water in test tubes. The tubes were shaken and centrifuged (CENTRIBIO) at 100 RPM for 30 minutes and then at 3500 RPM for 5 minutes.

The 2.5 mL aliquots of the supernatant were collected and placed in crucibles for drying in an oven at 105˚ for 1 hour. The dried materials were weighed for calculation using Eq 4, where $m_1$ is the mass of the 2.5 mL aliquot after drying, and $m_0$ is the mass incorporated into 5 mL of distilled water.

$$Dispersibility(\%) = \left[\frac{m_1 * 2}{m_0}\right] * 100 \qquad \text{(Eq 4)}$$

## Antioxidant activity

**Total Antioxidant Capacity (TAC).** TAC was determined based on Morais et al. (2022) [16] with modifications. To each tube, 100 μL of a 40 mM ammonium molybdate-sulfuric acid stock solution, 100 μL of 280 mM sodium phosphate, 100 μL of the samples at a concentration of 1 mg/mL, and 1.0 mL of distilled water were added. Then, the tubes were shaken and placed in a water bath (QUIMIS, Mod. Q334M-28) at 100˚ for 90 minutes to read the absorbances by spectrophotometry (Bioespectro UV-VIS, Mod. SP-220) at 695 nm.

The experiment was performed in triplicate, using the astaxanthin-pigmented oil as a control. The antioxidant activity was expressed in milligrams of ascorbic acid per gram of sample (mg AA/g sample). The standard curve was constructed using different concentrations of ascorbic acid (25–250 mg/g). It is worth mentioning that the encapsulating agents were previously evaluated for antioxidant activity and showed no activity.

**Iron ion chelation.** It was performed according to the methodology described by Lima et al. (2010) [29] with modifications. Each sample was diluted in distilled water (0.5 mg/mL), and to this aliquot were added 546 μL of distilled water, 30 μL of iron chloride (FeCl2), and 60 μL of ferrozine (3-(2-pyridyl)-5,6-bis(4-phenylsulfonic acid)-1,2,4-triazine). The mixture was stirred, and after 10 minutes in a water bath (QUIMIS, Mod. Q334M-28), the reading was performed in spectrophotometry (Bioespectro UV-VIS, Mod. SP-220) at 562 nm. The tests were performed in triplicate, and EDTA (200 mg/mL) was used as standard. The control activity was considered 100%, and the samples' chelating activity (% CA) was calculated according to Eq 5. It is worth mentioning that the encapsulating agents were previously evaluated for antioxidant activity and did not show any activity.

$$CA(\%) = 100 - \left[\frac{Ac - At}{Ac}\right] * 100 \qquad \text{(Eq 5)}$$

Ac: control absorbance; At: test absorbance (sample)

**Statistical analyses.** The results were expressed as mean and standard deviation and analyzed using STATISTICA 64® versão 12 (StatSotf, Inc.–Tulsa, OK/USA) software. Initially, the normality of data distribution was assessed using the Shapiro-Wilk test. Data that

presented normal distribution were analyzed using parametric tests using ANOVA followed by Tukey's post-test, and a p-value < 0.05 was adopted for statistically significant differences.

## Results and discussion

### Characterization of astaxanthin-pigmented oil

The astaxanthin-pigmented oil yielded 255.3 (4.0) mL, and the quantification of astaxanthin was 13.96 (1.24) μg/g of oil. According to the literature, Silva et al. (2021) [11] obtained 27.48 μg of astaxanthin/g by extracting with soybean oil. A similar result was obtained by Sachindra & Bhaskar (2008) [30], who also used soybean oil to promote extraction and obtained 24.8 μg/g of oil. While Liu et al. (2021) [31] extracted the pigment using an organic solvent and obtained 19.20 μg/g. Phadtare et al. (2021) [5] extracted oil from shrimp residue using Soxhlet and obtained 24.03 μg of astaxanthin/g.

Therefore, it is possible to note that the astaxanthin content present in the pigmented oil was lower compared to studies published in the literature, and this result can probably be justified by the fact that astaxanthin content varies in the diet of farmed shrimp since it is provided through feed supplementation [32]. Astaxanthin, as other carotenoids, cannot be synthesized by animals and is obtained through dietary intake [33]. The main application of astaxanthin is as an aquaculture feed supplement to improve coloration and fertility and can represent up to 15–20% of the total feed cost [34]. Auerswald & Gäde (2008) [35] observed that shrimp fed a diet containing astaxanthin showed a 20 to 41% increase in astaxanthin content after 30 days compared to those fed a diet without astaxanthin.

### Characterization of nanoparticles

**Colorimetry.** The formulations developed were astaxanthin-pigmented oil in porcine gelatin (EAG) (Fig 1A), astaxanthin-pigmented oil in porcine gelatin and soy protein in a 2:2 w/w ratio (EAGS 2:2) (Fig 1B), and astaxanthin-pigmented oil in porcine gelatin and soy protein in a 3:1 w/w ratio (EAGS 3:1) (Fig 1C). The colors of each encapsulated product were evaluated based on the $L^*$ $a^*$ $b^*$ indexes of the International Commission on Illumination (CIE) (Fig 1).

From the results obtained for the $L^*$ index (Table 1), it is possible to observe that the encapsulated EAGS 2:2, containing the highest soy protein content, presented a higher average for luminosity ($L^*$) compared to the others, differing only from EAGS 3:1 (p < 0.05). This is possibly due to the white and opaque coloration of the soy protein. Regarding the $a^*$ index, all results indicated the presence of green coloration, which, according to the literature, is related

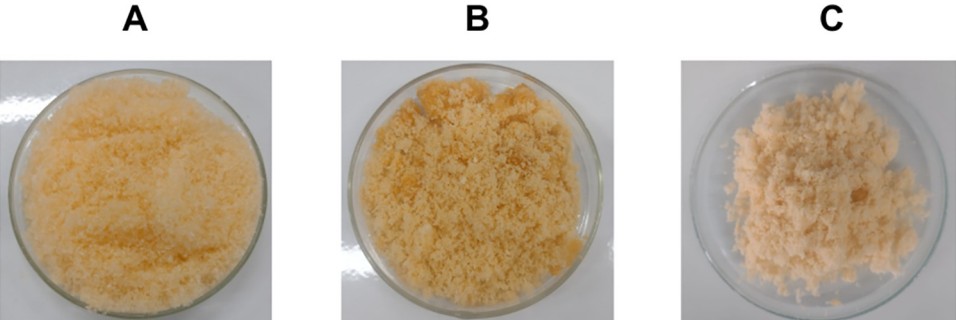

**Fig 1. Powder encapsulates containing astaxanthin-pigmented oil.** A: Astaxanthin-pigmented oil in porcine gelatin (EAG). B: Astaxanthin-pigmented oil in porcine gelatin and soy protein in a 2:2 w/w ratio (EAGS 2:2). C: Astaxanthin-pigmented oil in porcine gelatin and soy protein in a 3:1 w/w ratio (EAGS 3:1).

**Table 1. L* a* b* indexes obtained for the different encapsulations evaluated.**

| Parameter | EAG | EAGS 2:2 | EAGS 3:1 |
|---|---|---|---|
| L* | 37.50 (2.39)[a] | 39.55 (2.53)[a] | 29.12 (0.34)[b] |
| a* | -2.73 (1.27)[a] | -3.42 (1.38)[a] | -0.39 (0.41)[a] |
| b* | 7.99 (2.46)[a] | 7.84 (2.08)[a] | 9.43 (1.66)[a] |

Mean (standard deviation), n = 3.

According to ANOVA and Tukey's post-test (p < 0.05), equal letters in the same row mean no significant difference. EAG: astaxanthin-pigmented oil in porcine gelatin; EAGS 2:2: astaxanthin-pigmented oil in porcine gelatin and soy protein in a 2:2 w/w ratio; EAGS 3:1: astaxanthin-pigmented oil in porcine gelatin and soy protein in a 3:1 w/w ratio.

to the presence of porcine gelatin in the encapsulated products [36]. The results obtained for the b* index indicated the influence of the yellow coloration in all the encapsulated products, which, together with the green color, suggests the predominance of a resulting orange coloration characteristic of the astaxanthin present in the pigmented oil. Therefore, it is possible to state that the encapsulated products presented a desirable visual appearance, enabling future research to evaluate their use as a colorant in food products.

**Scanning Electron Microscopy (SEM).** According to the micrographs obtained (Fig 2), it was possible to verify that the encapsulated EAG (Fig 2A), EAGS 2:2 (Fig 2B), and EAGS 3:1 (Fig 2C) presented particles with a spherical shape, smooth surface, without depressions or cracks, with several agglomerations and physical size between 150 and 190 nm. The presence of a smooth surface suggests that the wall materials promoted the protection of the active ingredient [20].

Castro et al. (2020) [14] encapsulated buriti oil with porcine gelatin and a combination of porcine gelatin and sodium alginate, and Lira et al. (2020) [15] encapsulated quinoa oil with porcine gelatin and a combination of porcine gelatin and whey protein isolate. In both studies, the O/W emulsification technique was used. The particles observed in the SEM had a spherical shape and no cracks, indicating no oil leakage, data that corroborates the results of the present study. Prasad Reddy et al. (2019) [37] encapsulated coffee bean oil by nano spray drying using whey protein isolate, and the particles also had a spherical shape and a smooth surface.

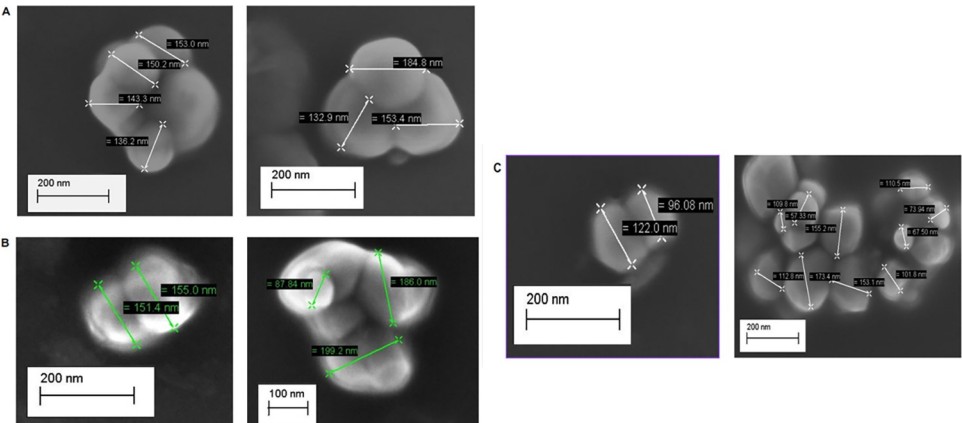

**Fig 2. SEM micrographs of powder particles dispersed in acetone, obtained by O/W emulsification.** A: Encapsulated containing astaxanthin-pigmented oil based on porcine gelatin (EAG) with magnitude 80.00KX. B: Encapsulated containing astaxanthin-pigmented oil based on porcine gelatin and soy protein in a 2:2 w/w ratio (EAGS 2:2) with magnitude 80.00KX. C: Encapsulated containing porcine gelatin-pigmented oil and soy protein in a 3:1 w/w ratio with magnitude 80.00KX (EAGS 3:1).

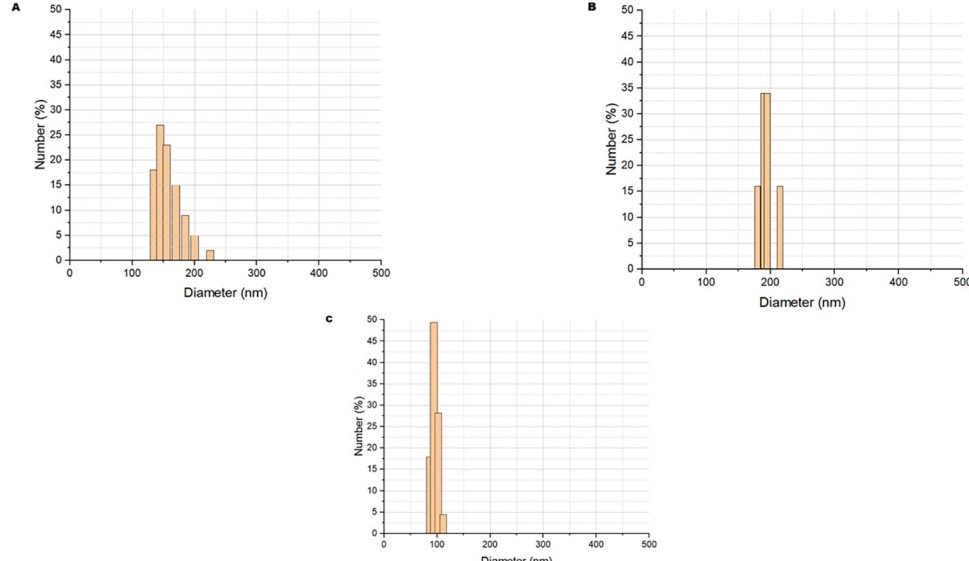

**Fig 3. Particle size distribution of powders by DLS obtained by O/W emulsification and subjected to crosslinking to reduce agglomeration.** A: Encapsulated containing astaxanthin-pigmented oil based on porcine gelatin (EAG). B: Encapsulated containing astaxanthin-pigmented oil based on porcine gelatin and soy protein at a 2:2 w/w ratio (EAGS 2:2). C: Encapsulated containing porcine gelatin-pigmented oil and soy protein at a 3:1 w/w ratio (EAGS 3:1).

**Dynamic Light Scattering (DLS).** The results (Fig 3) indicated unimodal particle size distribution obtained for each encapsulated. For EAG (Fig 3A), the average size was 159.68 (14.42) nm, and the Polydispersity Index (PDI) was 0.08. EAGS 2:2 (Fig 3B) showed a tendency towards particle agglomeration and slightly higher values in the size distribution (180–210 nm), with an average of 192.72 (10.44) nm and a PDI of 0.32. Regarding EAGS 3:1 (Fig 3C), it was observed to be encapsulated with the most minor diameter, with an average of 95.41 and a PDI of 0.23 (17.83) nm.

The Polydispersity Index (PDI) reveals the diameter distribution's amplitude and the particles' degree of homogeneity. Therefore, it is essential to complement the evaluation of particle size. A PDI value from 0 to 0.4 indicates that the particles have a homogeneous size range [22]. Thus, it is noted that all the encapsulates evaluated presented a homogeneous size distribution since the PDI in all samples was below 0.4.

Gulzar et al. (2022) [38] evaluated the average size of particles obtained by O/W emulsification using chitosan and/or tripolyphosphate, Tween 80, and shrimp waste oil in different proportions of the wall material. They found results between 204.32 and 278.11 nm. Just like Yang et al. (2022) [39] in the encapsulation of *H. pluviallis* extract containing 10% astaxanthin using different wall materials (milk protein, cornstarch, lactoferrin, soy protein, and sodium caseinate containing 95% protein) through the O/W emulsification technique followed by spray-drying and obtained particles with diameters in the range of 246.3–442.7 nm.

Tirado et al. (2019) [40] evaluated the encapsulation through the O/W emulsification technique using commercial astaxanthin, ethylcellulose, and different concentrations of the surfactant Tween 80 and related the amount of encapsulating agent present, emulsification power, and viscosity of the material with the particle size obtained. They obtained particle size results between 161 nm and 733 nm. As noted, the choice of encapsulating agents and encapsulation technique directly influence the determination of particle size distribution, implying the final purpose of use of the encapsulated products Martínez-Álvarez et al (2020) [41]. Particle size distribution can improve solubility, bioaccessibility, and functional properties since smaller

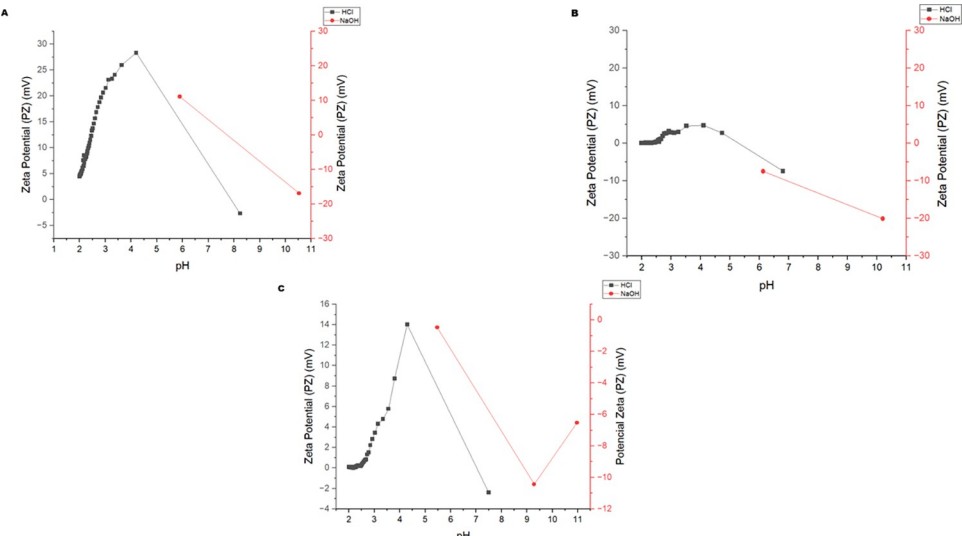

**Fig 4. Zeta potential of the encapsulates obtained by O/W emulsification at different pH.** A. EAG: oil pigmented with astaxanthin in porcine gelatin; B. EAGS 2:2: oil pigmented with astaxanthin in porcine gelatin and soy protein in a 2:2 w/w ratio; C. EAGS 3:1: oil pigmented with astaxanthin in porcine gelatin and soy protein in a 3:1 w/w ratio.

particle sizes present better performance and more satisfactory results in future applications, such as food matrices or pharmaceutical products [16, 42].

**Zeta potential.** The Zeta potential (mV) values are presented concerning a pH range of 1 to 11 (Fig 4). pH is one of the main factors influencing Zeta potential since particles with average values above ±30 mV are highly stable, while moderately stable ones are on average between ±10–20 mV, and highly unstable ones when they present an average charge between ±0-10mV [43].

EAG (Fig 4A) showed stability at pH around 4.5, as it presented Zeta potential values above +28 mV, indicating the presence of positive charges on the surface of the particles due to the pH below the isoelectric point (pH = 7.0) of porcine gelatin [16]. It is worth noting that this can influence the definition of the product type in which the particles can be used [14]. Furthermore, it is noteworthy that particles with a predominance of positive charge on the surface facilitated interaction with phospholipid membranes, favoring controlled release [15].

The EAGS 2:2 encapsulated (Fig 4B) at pH 10 presented charges <-20 mV, indicating moderate stability. The presence of negative charges on the surface suggests that the pH is above the isoelectric point of soy protein (pH = 4 to 5). Soy protein presents greater stability when the pH moves away from its isoelectric point. Thus, limited use in acidic products is observed [44, 45].

For EAGS 3:1 (Fig 4C), it was observed that at pH around 4.0, the system reached a Zeta potential above +14 mV, demonstrating the influence of porcine gelatin on the surface of the particles since the charges have a positive predominance when the pH is below 7.0 [14] This result indicates that the particles have moderate stability, which may favor agglomeration [43].

The Zeta potential results of the EAG encapsulated varied according to the pH change, presenting positive and negative charges corresponding to the moderate stability range due to the amphoteric character of porcine gelatin [28]. Therefore, the encapsulated matter stood out regarding the stability of surface charges compared to the others evaluated. The higher the absolute value of the Zeta potential, the greater the electrostatic force of repulsion on the surface of the particles, preventing agglomeration [46].

Ge et al. (2023) [42] promoted the encapsulation of commercial astaxanthin in isolated soy protein with or without sodium alginate under different storage, light, and temperature conditions. They found negative values for the Zeta potential (>-30 mV); thus, the particles maintained the repulsion of electrostatic charges, helping to prevent agglomeration. Gu et al. (2023) [47] used glycerol, β-cyclodextrin, and various emulsifiers to promote the encapsulation of commercial astaxanthin oil by high-pressure homogenization. Thus, the Zeta potential of the nanoparticles was equal to 26.16 (0.87) mV, which means moderate stability.

## Fourier Transform Infrared Spectroscopy (FTIR)

The spectrum corresponding to the oil pigmented with astaxanthin (Fig 5) showed a vibrational band at 3011 cm$^{-1}$, highlighting the presence of an O-H bond. Two bands evidenced at 2922 cm-1 and 2851 cm$^{-1}$ related to the symmetric and asymmetric methylene bonds (-CH$_2$) were also present, as was a vibrational band in the region of 1742 cm$^{-1}$ characteristic of C = O and C = C double bonds present in unsaturated oils (Lira et al., 2020).

In the spectrum of the surfactant Tween 20 (Fig 5), a vibrational band was observed at 3492 cm$^{-1}$ related to the presence of an O-H bond, two bands at 2926 cm$^{-1}$ and 2864 cm$^{-1}$ referring to C-H bonds, another at 1738 cm$^{-1}$ related to the C = O bond, and a sharp vibrational band at 1091 cm$^{-1}$ that characterizes the stretching of double bonds (C = C and C = O) [16].

For porcine gelatin (Fig 5), a vibrational band is observed in the region of 3281 cm$^{-1}$, referring to the O-H bond; a band at 1626 cm$^{-1}$ indicating the presence of the amide I bond (C = O), and also at 1526 cm$^{-1}$, indicating the presence of acyclic secondary amide (N-H). The spectrum of the isolated soy protein (Fig 5B) showed bands in the regions 3274 cm$^{-1}$ and 3050 cm$^{-1}$ referring to O-H bonds, other bands in the regions 2957 cm$^{-1}$, 2926 cm$^{-1}$ and 2864 cm$^{-1}$ that reveal symmetrical and asymmetrical groups of methylene -CH$_2$ bonds, and accentuated vibration at 1632 cm$^{-1}$ that indicates C = O bond of amide I and also a band at 1526 cm$^{-1}$ referring to the bond of the cyclic secondary amide (N-H) [48].

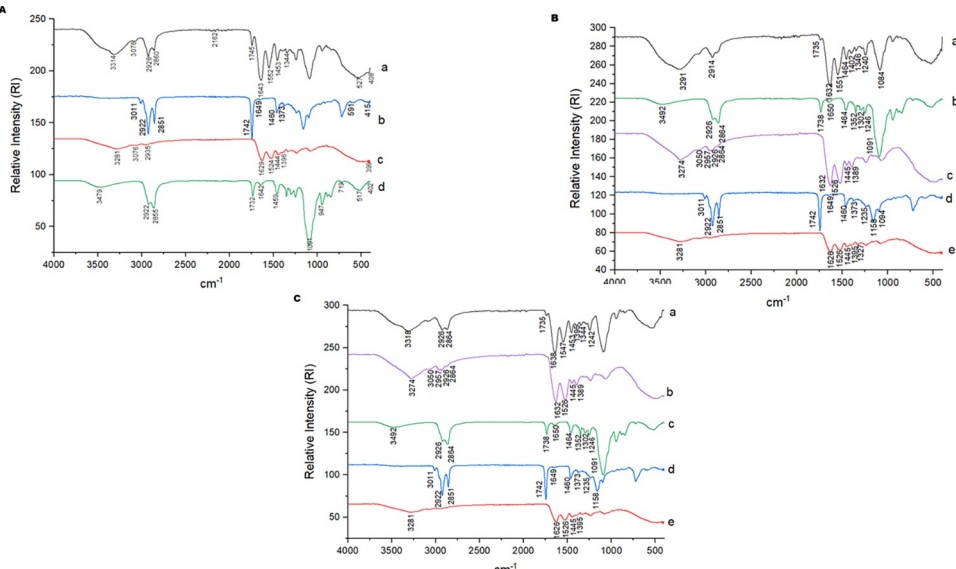

**Fig 5. FTIR of the encapsulates obtained by O/W emulsification.** A—EAG: oil pigmented with astaxanthin in porcine gelatin, a: EAG; b: Pigmented oil with astaxanthin (POA); c: Porcine gelatin; d: Tween 20; B—EAGS 2:2: oil pigmented with astaxanthin in porcine gelatin and soy protein in a 2:2 w/w ratio, a: EAGS 2:2; b: Tween 20; c: Soy Protein Isolate; d: Pigmented oil with astaxanthin (POA); e: Porcine Gelatin; C—EAGS 3:1: oil pigmented with astaxanthin in porcine gelatin and soy protein in a 3:1 w/w ratio, a: EAGS 3:1; b: Tween 20; c: Soy Protein Isolate; d: Pigmented oil with astaxanthin (POA); e: Porcine gelatin.

In the spectrum of the encapsulated EAG (Fig 5A), the displacement of the vibrational bands related to the O-H and C-H stretching that characterize the presence of gelatin (3314 cm⁻¹, 3078 cm⁻¹) is observed, in addition to the bands 2928 cm⁻¹ and 2860 cm⁻¹, which suggest the presence of oil pigmented with astaxanthin. The stretching band of the C = O double bond of esters (1745 cm⁻¹) [49] characteristic of lipids was visualized with slight vibration. The bands that characterize the nitrogen compounds were displaced in the encapsulated, as observed at 1643 cm⁻¹ and 1522 cm⁻¹ (N-H).

For the EAGS 2:2 encapsulated (Fig 5B), it was possible to observe new bands at 3291 cm⁻¹, representing O-H bonds [15], and at 1735 cm⁻¹, characteristic of stretching of C = O double bonds of esters present in unsaturated lipids [50], suggesting the chemical interaction between the wall materials and pigmenting oil. The band at 2914 cm⁻¹ was attenuated and appeared displaced, related to the pigmented oil [16]. Cyclic secondary amide N-H and amide I C = O bond [15] present in soybean protein and porcine gelatin were revealed by the vibrational bands at 1632 cm⁻¹ and 1551 cm⁻¹.

In the EAGS 3:1 encapsulated (Fig 5C), new bands were observed at 3318 cm-1 referring to the O-H bond [15], and the band related to the C = O stretching (1735 cm⁻¹) present in esters [48], which suggest chemical interactions of the pigmented oil with the encapsulating agents present. The vibrational bands observed at 1638 cm⁻¹ and 1547 cm⁻¹, referring to the cyclic secondary amide (N-H) and amide I (C = O) bond [15], reveal the interaction of the pigmented oil with soybean protein and porcine gelatin.

Lira et al. (2020) [15], using the porcine gelatin as wall material, also observed vibrational band shifts in the encapsulated material obtained, suggesting chemical interaction with the quinoa oil used as the core. Thus, these results obtained for the encapsulated materials in the present study indicate a chemical interaction between the encapsulating agents and pigmented oil.

## X-ray diffraction (XRD)

The diffractogram of isolated soy protein (Fig 6A) shows a peak between noise at 2θ equal to 9.0˚ and another at an angle of 19.5˚. This same pattern was verified in the literature [51], which is related to the α-helix and β-sheet structure of soy protein, indicating the presence of a semicrystalline structure in the material.

Porcine gelatin (Fig 6B) also presented a smaller peak between noises located at 2θ equal to 8.0˚ and a larger one at an angle of 20.0˚, corresponding to the triple helix structure of collagen [22]. Castro et al., (2020) [14] also reported the presence of peaks between noises and a semicrystalline structure for porcine gelatin. Mosleh et al. (2021) [52] carried out a study on the structure of powdered gelatin. They obtained a similar diffractogram in the XRD analysis, with a peak of intensity relative to the triple helix content of collagen present in porcine gelatin.

Regarding EAG, EAGS 2:2, and EAGS 3:1 (Fig 6C–6E), the presence of a semicrystalline structure and attenuation of a minor peak between noises of the encapsulating agents at 2θ equal to 9.0˚ were observed, and the presence of a more prominent peak between noises at the 2θ angle equal to 19.5˚ in the EAGS 2:2 encapsulate. The same behavior was evidenced by Lira et al. (2020) [15], indicating the increase in the peak intensity in the same region present in the encapsulates containing porcine gelatin. Therefore, the diffractograms of the encapsulate suggest the interaction of the encapsulating agent's porcine gelatin and isolated soy protein with the content of oil pigmented with astaxanthin promoted by the O/W emulsification technique.

## Incorporation efficiency (IE)

Incorporation efficiency (IE) is considered optimal when results reach percentages >80% [53]. To obtain higher EI values, it is necessary to use materials that have emulsifying and protective

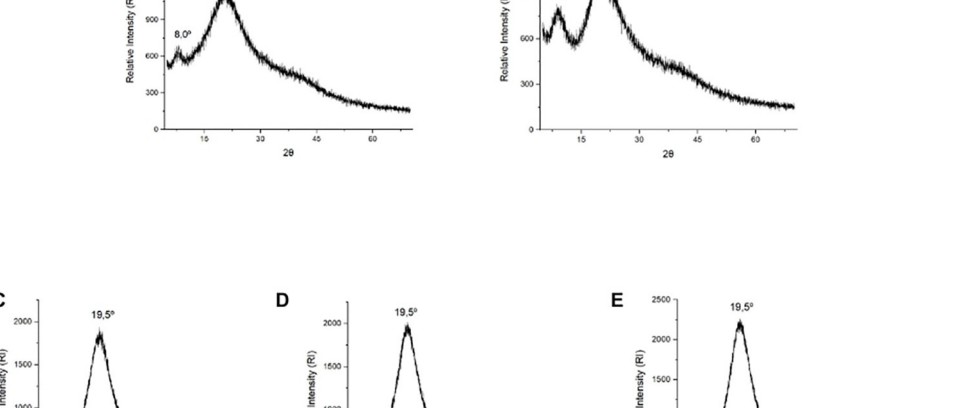

**Fig 6. X-ray diffractograms of the materials and encapsulates obtained by O/W emulsification.** A: Soy Protein. B: Porcine Gelatin. C: Encapsulate containing astaxanthin-pigmented oil based on porcine gelatin (EAG). D: Encapsulate containing astaxanthin-pigmented oil based on porcine gelatin and soy protein in a 2:2 w/w ratio (EAGS 2:2). E: Encapsulate containing porcine gelatin-pigmented oil and soy protein in a 3:1 w/w ratio (EAGS 3:1).

film-forming properties that can quickly coat the droplets formed and reduce the diffusion of the retained oil to the surface of the particles obtained [2]. The high EI means that the oil pigmented with astaxanthin is less exposed to the environment and consequently provides a better protective effect [39].

Based on the results (Table 2), all encapsulates exceeded 80% of incorporation efficiency, representing a high efficiency in incorporating the pigmented oil. The encapsulates with a higher proportion of porcine gelatin (EAG and EAGS 3:1) as encapsulating agent obtained higher percentages compared to the encapsulate with a lower proportion of gelatin (EAGS 2:2), confirming the behavior in previous studies [15, 16, 36] of better performance of porcine gelatin in encapsulation by O/W emulsification.

Volić et al (2018) [54] observed that combining soy protein and alginate increased the encapsulation efficiency of essential oil. Still, higher proportions of soy protein indicate that the incorporation efficiency may decrease probably because the protein molecules may partially occupy the free volume within the polymeric matrix available for the oil.

**Table 2. Incorporation efficiency (IE) of EAG, EAGS 2:2, and EAGS 3:1 encapsulates.**

| Encapsulated | IE (%) |
|---|---|
| EAG | 89.20 (0.31)[a] |
| EAGS 2:2 | 87.60 (0.13)[b] |
| EAGS 3:1 | 89.10 (0.30)[a] |

Mean (standard deviation), n = 3.

According to ANOVA and Tukey's post-test (p<0.05), equal letters in the same column mean no significant difference.

EAG: astaxanthin-pigmented oil in porcine gelatin; EAGS 2:2: astaxanthin-pigmented oil in porcine gelatin and soy protein in a 2:2 w/w ratio; EAGS 3:1: astaxanthin-pigmented oil in porcine gelatin and soy protein in a 3:1 w/w ratio.

Gulzar et al. (2022) [38] promoted the encapsulation by O/W emulsification of shrimp waste oil in different proportions and chitosan-tripolyphosphate and obtained results between 32.34% (4.89) to 67.54% (3.79). Another study with astaxanthin-rich shrimp oil encapsulated by O/W emulsification using Persian gum and isolated milk protein reached averages of 42.95% to 49.85% for incorporation efficiency [25], results lower than the present study.

Panagiotakopoulos et al. (2023) [8] encapsulated astaxanthin from shrimp waste solubilized in different oils in gum arabic and soybean lecithin using ultrasound and achieved between 28% (day 6) and 66% (day 1) of incorporation efficiency. The encapsulation performed by Gomez-Estaca et al. (2016) [55] of shrimp waste oil in cashew gum and porcine gelatin by O/W emulsification and coacervation reached 59.9% (0.01). Montero et al. (2016) [2] achieved 94% (1.00) efficiency when encapsulating lipid extract from shrimp waste in maltodextrin and/or gum arabic by emulsification and sonication. Takeungwongtrakul et al. (2015) [20] encapsulated shrimp hepatopancreas oil in concentrated whey protein by O/W emulsification, using sodium caseinate and glucose syrup, resulting in an incorporation efficiency equal to 86.31% (1.88), corroborating the data of the present study.

The incorporation efficiency directly relates to the conditions and technique used during encapsulation [15]. The results obtained superior performance to that found in the literature, especially for emulsions using only gelatin and the combination of encapsulating agents with a higher proportion of porcine gelatin (EAG and EAGS 3:1). Therefore, it is suggested that porcine gelatin tends to form more stable particles during O/W emulsification, probably promoting greater chemical interaction between the nonpolar chain of the pigmented oil and nonpolar amino acids present in the encapsulating agent [56, 57]. Nevertheless, soy protein isolate also demonstrated steric stabilization abilities in emulsification, promoting interaction with the pigmented oil [58]. It is reiterated that high incorporation efficiency values are desirable to facilitate the use of particles containing bioactive compounds in the development of functional foods [59].

Since astaxanthin was extracted from the pigmented oil, the encapsulated products have the molecule as part of their composition. Thus, the incorporation of the astaxanthin molecule was calculated indirectly in the encapsulated products (Table 3).

There was no significant difference in the results of astaxanthin incorporation in the encapsulated products, which demonstrates that the encapsulation provided equal retention of astaxanthin. As observed, such results may be due to the chemical interactions of the pigmented oil with the wall materials used and the surfactant (Tween 20) in the preparation of the emulsions in different proportions of encapsulating agents, in addition to the conditions in the freeze-drying process to provide particles with more stable systems [25]. Therefore, the greater the

**Table 3. Incorporation efficiency (IE) of astaxanthin in EAG, EAGS 2:2 and EAGS 3:1 encapsulates.**

| Encapsulated | IE (%) |
|---|---|
| EAG | 68.40 (0.24)[a] |
| EAGS 2:2 | 72.50 (5.65)[a] |
| EAGS 3:1 | 67.90 (0.23)[a] |

Mean (standard deviation), n = 3.

According to ANOVA and Tukey's post-test ($p < 0.05$), equal letters in the same column mean no significant difference.

EAG: astaxanthin-pigmented oil in porcine gelatin; EAGS 2:2: astaxanthin-pigmented oil in porcine gelatin and soy protein in a 2:2 w/w ratio; EAGS 3:1: astaxanthin-pigmented oil in porcine gelatin and soy protein in a 3:1 w/w ratio.

number of chemical interactions between the materials present in the system, the greater the retention of the bioactive compound after encapsulation [20].

The shrimp waste oil nanoparticles in different proportions of chitosan-tripolyphosphate obtained by Gulzar et al. (2022) [38] achieved astaxanthin incorporation capacity between 15.67% (0.76) and 23.69% (0.74). The krill oil microcapsules in gum arabic obtained by Ortiz Sánchez et al. (2021) [60] revealed astaxanthin retention between 66.77% and 86.18%, values closer to those obtained in the present study.

## Water dispersibility assay

Astaxanthin-pigmented oil's low stability and water insolubility reduce bioavailability and limit applicability, as it prevents incorporation into food matrices. Developing a strategy to expand the potential use of pigmented oil is essential to promote its utilization and functional properties [46]. In the present study, the encapsulated products presented water dispersibility (%) with values of 56.25% (3.04), 52.21% (4.78), and 58.07% (3.53) for EAG, EAGS 2:2, and EAGS 3:1, respectively (Fig 7).

Statistical analysis showed no significant difference ($p > 0.05$) between the water dispersibility results of the encapsulated oil. On the other hand, the dispersibility of the pigmented oil was only 0.19% (0.19), differing statistically from the encapsulated oil ($p < 0.05$), indicating that the nanoencapsulation added a characteristic not present in the crude oil, and this can be attributed to the particle diameters obtained, which promoted a greater surface area for interaction with water.

Furthermore, both porcine gelatin and isolated soy protein were shown to be efficient in promoting the water solubility of the pigmented oil, favoring an increase in water dispersibility by approximately 300 times. Thus, proving to be viable options for application and resulting in the addition of a characteristic not present in oils [14–16, 42, 61–65].

Porcine gelatin is hydrophilic, providing increased solubility of the encapsulated pigmented oil. In an O/W emulsion system, soy protein can form viscoelastic interphase layers on the surface of the oil particles, which prevent aggregation due to electrostatic repulsion energy [66]. On the other hand, the application of soy proteins in encapsulation is still restricted due to its limited solubility in water and high viscosity [67]. Therefore, it becomes a challenge to establish a proportion of porcine gelatin and soy protein that presents uniform particles with high dispersibility. From the results of the present study, it is possible to infer that the proportions of encapsulating agents investigated favored the water dispersibility of the systems containing soy protein (EAGS 2:2 and EAGS 3:1).

Bassijeh et al. (2020) [25] evaluated the solubility of particles based on Persian gum and milk protein containing astaxanthin-rich shrimp oil produced by multilayer emulsification and obtained a dispersibility percentage above 94%. In contrast, Gomez-Estaca et al. (2016) [55], for particles based on cashew gum containing astaxanthin-rich krill oil, obtained 28.6 (4.7)% solubility in water.

## Antioxidant potential

The results (Fig 8A) show that the nanoencapsulation process preserved (EAG) or enhanced (EAGS 2:2 and EAGS 3:1) the total antioxidant capacity of the pigmented oil. These results can be explained by the dispersibility in water and smaller particle sizes obtained [16, 68]. Thus, it can be inferred that the pigmented oil's antioxidant activity increased due to the smaller particle diameter and, consequently, the increase in the contact surface for chemical interaction [28]. This behavior was mainly observed in the EAG 3:1 encapsulated, which presented a smaller particle size and more significant antioxidant activity.

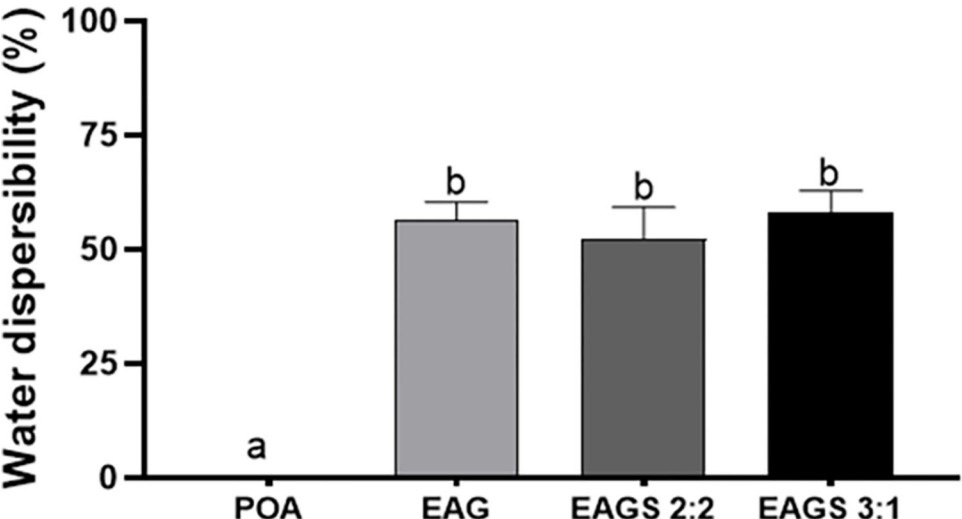

**Fig 7. Dispersibility in water of the pigmented oil with astaxanthin and encapsulates evaluated.** POA: pigmented oil with astaxanthin; EAG: pigmented oil with astaxanthin in porcine gelatin; EAGS 2:2: pigmented oil with astaxanthin in porcine gelatin and soy protein in a 2:2 w/w ratio; EAGS 3:1: pigmented oil with astaxanthin in porcine gelatin and soy protein in a 3:1 w/w ratio. ANOVA and Tukey's post-test ($p < 0.05$).

The pigmented oil and the encapsulated oil contain astaxanthin in their composition, in which the chemical chain with double hydroxyl has antioxidant capacity. The nanometric size of the EAGS 3:1 encapsulated oil may have contributed to doubling the antioxidant capacity compared to the non-encapsulated pigmented oil. It is essential to highlight that 1 mg/mL of the EAG, EAGS 2:2, and EAGS 3:3 encapsulated contains 0.446, 0.438, and 0.445 mg/mL of the pigmented oil with astaxanthin, which reinforces the enhancement of the antioxidant activity of the same in the nanoparticles.

Furthermore, combining the functionalities of both wall materials may have influenced the improvement of the antioxidant activity due to the chemical interactions in the system [69]. The study by Morais et al. (2022) [16] investigated the Total Antioxidant Capacity for crude buriti oil and the same encapsulated with porcine gelatin and obtained values equal to 14.60 mg AA/g (1.63) and 48.34 (3.71) mg AA/g, respectively, relating the best antioxidant activity when the oil is encapsulated. A situation similar to the results found for the encapsulates obtained in the present study.

The results obtained for the iron ion chelating activity (Fig 8B) indicated that the encapsulated samples with a higher proportion of gelatin (EAG and EAGS 3:1) enhanced the iron ion chelating activity by approximately three times. At the same time, the sample with a higher proportion of isolated soy protein (EAGS 2:2) preserved the antioxidant effect compared to the non-encapsulated pigmented oil. Porcine gelatin has essential amino acids in its composition, such as proline, glycine, and arginine, which also have antioxidant functions, and due to the chemical interactions with astaxanthin present in the oil, they may have helped to increase the iron chelating [70].

No results were found in the literature for this analysis involving iron ions using astaxanthin-pigmented oil or encapsulated products containing astaxanthin-pigmented oil. However, Suganya & Asheeba (2015) [71] evaluated residues from crab species containing

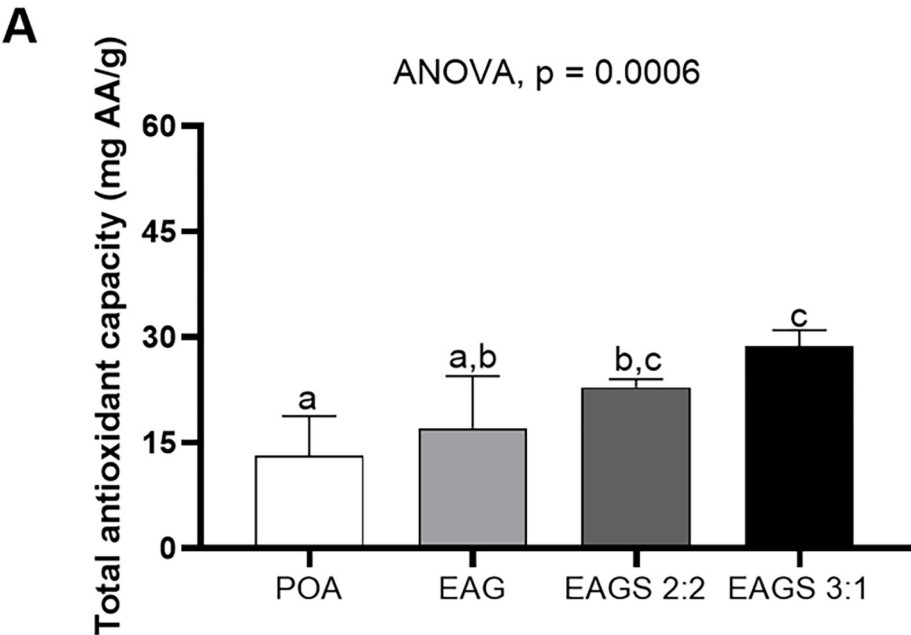

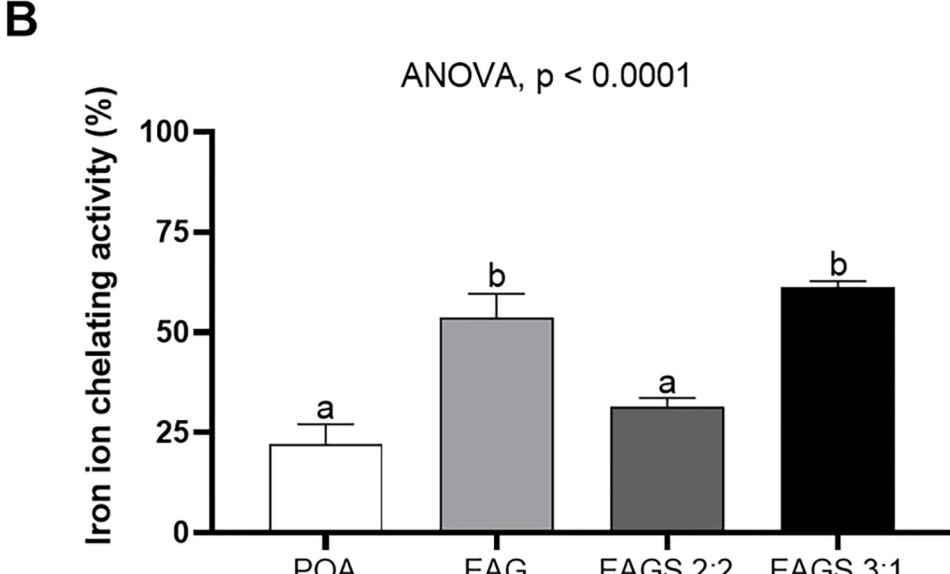

**Fig 8. Antioxidant potential of the astaxanthin-pigmented oil and the evaluated encapsulates.** A. Total Antioxidant Capacity. B. iron ion chelating activity. POA: pigmented oil with astaxanthin; EAG: pigmented oil with astaxanthin in porcine gelatin; EAGS 2:2: pigmented oil with astaxanthin in porcine gelatin and soy protein in a 2:2 w/w ratio; EAGS 3:1: pigmented oil with astaxanthin in porcine gelatin and soy protein in a 3:1 w/w ratio. ANOVA and Tukey's post-test (p < 0.05).

astaxanthin and obtained results for iron ion chelation between 40% and 52%, similar to the present study.

The results indicated that the encapsulated products can form complexes with iron ions, making them inactive and preserving them from oxidation [72]. The increase in solubility and nanometric particle size observed in the particle characterization results influenced the

antioxidant activity of the second oxidative pathway, corroborating the stabilization of the system containing astaxanthin [73, 74].

The antioxidant activity in both analyses (CAT and iron ion chelation) demonstrated preservation or enhancement when comparing the encapsulated and non-encapsulated pigmented oil. The study revealed that the astaxanthin present in the oil and the encapsulated products can act as an antioxidant by eliminating reactive oxygen species and preventing iron ion chelation, an essential characteristic for evaluating the future application in the food matrix.

## Conclusion

The O/W emulsification technique using gelatin and/or soy protein has proven effective in encapsulating astaxanthin-pigmented oil, reaching nanometric particle size, and promoting water solubility, preservation, and/or enhancement of antioxidant activity. The encapsulated products, emphasizing the EAGS 3:1 formulation, have potential for application in food matrix due to their physicochemical characteristics, proving to be a promising and sustainable alternative for shrimp waste.

## Supporting information

**S1 File.**
(XLSX)

## Acknowledgments

We thank Faif's Mariculture for kindly providing the shrimp waste and the Laboratory of Multifunctional Materials and Nanocomposites (Lammen) of the School of Science and Technology of UFRN.

## Author Contributions

**Conceptualization:** Renata Nayane Fernandes dos Santos, Cristiane Fernandes de Assis.

**Data curation:** Rafael da Silva Fernandes.

**Formal analysis:** Renata Nayane Fernandes dos Santos.

**Investigation:** Karla Suzanne Florentino da Silva Chaves Damasceno, Cristiane Fernandes de Assis.

**Methodology:** Kátia Nicolau Matsui, Francisco Canindé de Sousa Júnior, Karla Suzanne Florentino da Silva Chaves Damasceno.

**Project administration:** Thaís Souza Passos, Cristiane Fernandes de Assis.

**Resources:** Kátia Nicolau Matsui.

**Software:** Francisco Canindé de Sousa Júnior.

**Supervision:** Thaís Souza Passos, Karla Suzanne Florentino da Silva Chaves Damasceno, Cristiane Fernandes de Assis.

**Writing – original draft:** Renata Nayane Fernandes dos Santos.

**Writing – review & editing:** Thaís Souza Passos, Karla Suzanne Florentino da Silva Chaves Damasceno, Cristiane Fernandes de Assis.

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
