## [Decision Letter · Decision Letter 0]

4 Oct 2024

PONE-D-24-38479Effect of nanoencapsulation on the solubility and antioxidant activity of astaxanthin pigmented oil extracted from shrimp waste (Litopeneaus vannamei)PLOS ONE

Dear Dr. de Assis,

Thank you for submitting your manuscript to PLOS ONE. After careful consideration, we feel that it has merit but does not fully meet PLOS ONE’s publication criteria as it currently stands. Therefore, we invite you to submit a revised version of the manuscript that addresses the points raised during the review process.

We look forward to receiving your revised manuscript.

Kind regards,

Arumugam Muthuvel

Academic Editor

PLOS ONE

Journal Requirements:

3. Thank you for stating the following financial disclosure: “This study was partly financed by the

Coordenação de Aperfeiçoamento de Pessoal de Nível Superior - Brasil (CAPES) - Finance Code 001.”

Reviewers' comments:

Reviewer's Responses to Questions

**Comments to the Author**

1. Is the manuscript technically sound, and do the data support the conclusions?

Reviewer #1: Yes

Reviewer #2: Yes

2. Has the statistical analysis been performed appropriately and rigorously? 

Reviewer #1: Yes

Reviewer #2: Yes

3. Have the authors made all data underlying the findings in their manuscript fully available?

Reviewer #1: Yes

Reviewer #2: Yes

4. Is the manuscript presented in an intelligible fashion and written in standard English?

Reviewer #1: Yes

Reviewer #2: Yes

5. Review Comments to the Author

Reviewer #1: - The manuscript is based on the nanoencapsulation of astaxanthin with a concluding remark for usage as a food matrix

- If that being the finding, preliminary study for human consumption needs to be checked and that's a future prospect.

- In the materials and methods many parameters are studied, shell life of the nanoencapsulation needs to be discussed

- If administered orally how does the absorption take place in the gut of the animal

- The images Fig.3, Fig.4, Fig.5, Fig.6 needs to have higher resolution

Reviewer #2: A detailed study and analysis of the hypothesis and laboratory work have been observed. All the required characterzation and instrumenation have been utilized, and the required data have been presented in a scientific and readable manner.

6. PLOS authors have the option to publish the peer review history of their article (what does this mean?). If published, this will include your full peer review and any attached files.

Reviewer #1: No

Reviewer #2: No

---

## [Author Response · Author response to Decision Letter 0]

16 Oct 2024

RESPONSE TO REVIEWERS

Reviewer #1:- The manuscript is based on the nanoencapsulation of astaxanthin with a concluding remark for usage as a food matrix

R. Thank you for considering our manuscript. We hope the revised manuscript is suitable for publication in Plos One. 

- If that being the finding, preliminary study for human consumption needs to be checked and that's a future prospect.

R. We appreciate the comment. The initial objective of this work was to evaluate the encapsulation of pigmented oil using porcine gelatin and soy protein, characterizing these nanoparticles and evaluating some bioactive properties. As a next step of the research, toxicity tests will be performed to ensure the safety of the particles.

- In the materials and methods many parameters are studied, shell life of the nanoencapsulation needs to be discussed

R. This study evaluated whether the wall materials efficiently promoted the nanoencapsulation of astaxanthin-pigmented oil, and the effects of nanoencapsulation on the bioactive potential of buriti oil. The nanoparticles have not yet been added to the food matrix, as further studies are needed to achieve their application. Therefore, shelf-life tests will be performed in the future.

- If administered orally how does the absorption take place in the gut of the animal?

R. The investigation of digestion and absorption in vitro or in vivo was not the objective of this study. First, different wall materials were evaluated to find the most efficient one to promote the nanoencapsulation of the astaxanthin-pigmented oil through the O/W emulsification technique, to provide the oil with solubility in an aqueous matrix and preserve or enhance the antioxidant activity. Based on this study, in the future, the nanoparticles will be evaluated for toxicity and bioactive potential in vitro and in vivo, aiming to elucidate other questions and expand the knowledge about the nanoparticles produced. It is expected that the digestion of the nanoparticles will occur in the stomach, since protein encapsulating agents were used, with controlled release of the buriti oil containing the bioactive compounds.

- The images Fig.3, Fig.4, Fig.5, Fig.6 needs to have higher resolution.

R. We appreciate the comments and provide a better view of Figures 3, 4. 5 and 6.

Reviewer #2

A detailed study and analysis of the hypothesis and laboratory work have been observed. All the required characterzation and instrumenation have been utilized, and the required data have been presented in a scientific and readable manner.

R. Thank you for considering our manuscript. We hope the revised manuscript is suitable for publication in Plos One.

---

## [Editor Report · Decision Letter 1]

18 Oct 2024

Effect of nanoencapsulation on the solubility and antioxidant activity of astaxanthin pigmented oil extracted from shrimp waste (Litopeneaus vannamei)

PONE-D-24-38479R1

Dear Cristiane Fernandes de Assis

We’re pleased to inform you that your manuscript has been judged scientifically suitable for publication and will be formally accepted for publication once it meets all outstanding technical requirements.

Kind regards,

Arumugam Muthuvel

Academic Editor

PLOS ONE
---

## [Editor Report · Acceptance letter]

6 Nov 2024

PONE-D-24-38479R1 

PLOS ONE

Dear Dr. de Assis, 

I'm pleased to inform you that your manuscript has been deemed suitable for publication in PLOS ONE. Congratulations! Your manuscript is now being handed over to our production team.

Kind regards, 

on behalf of

Dr. Arumugam Muthuvel 

Academic Editor

PLOS ONE